# Family Self-Care in the Context of Intellectual Disabilities: Insights from a Qualitative Study in Portugal

**DOI:** 10.3390/healthcare13141705

**Published:** 2025-07-15

**Authors:** Teresa Dionísio Mestre, Manuel José Lopes, Ana Pedro Costa, Ermelinda Valente Caldeira

**Affiliations:** 1Health Department, Polytechnic Institute of Beja, 7800-111 Beja, Portugal; 2Comprehensive Health Research Center [CHRC], 7004-516 Évora, Portugal; mjl@uevora.pt (M.J.L.); anapedrocosta92@gmail.com (A.P.C.); ecaldeira@uevora.pt (E.V.C.); 3Health Department, University of Évora, 7004-811 Évora, Portugal; 4Local Health Unit of Lower Alentejo [ULSBA], 7800-849 Beja, Portugal

**Keywords:** family self-care, intellectual disability, family nursing, qualitative study, self-care, family-centered care

## Abstract

**Background/Objectives**: Family self-care (FSC) is increasingly recognized as a vital aspect of caregiving in pediatric chronic conditions. However, its development in families of children with intellectual disabilities (IDs) remains underexplored. This study aimed to examine how families construct and sustain FSC, and to identify factors that shape its development across four domains: physical, cognitive, psychosocial, and behavioral. **Methods**: A qualitative study was conducted using an abductive approach, combining inductive thematic analysis with a deductively applied theoretical framework. Semi-structured interviews were carried out with nine families of children with ID in southern Portugal. The children ranged in age from 4 to 15 years, and the parents were aged between 29 and 53 years. The data was analyzed using Bardin’s content analysis, supported by NVivo software, and organized according to the FSC framework. This study followed COREQ guidelines. **Results**: The families described a range of self-care strategies, including environmental adaptations, experiential learning, emotional regulation, and long-term planning. These practices were shaped by contextual factors such as access to healthcare, relationships with professionals, emotional support networks, and socioeconomic conditions. Four emergent conclusions illustrate how structural and relational dynamics influence FSC in daily caregiving. **Conclusions**: FSC is a dynamic, multidimensional process shaped by lived experience, family interactions, and systemic support. The findings support inclusive, family-centered care models and inform clinical practice, training, and policy in pediatric IDs.

## 1. Introduction

Caring for a child with an intellectual disability (ID) is a complex, long-term responsibility that profoundly affects the structure, dynamics, and functioning of a family as a whole [1]. Globally, the prevalence of ID is estimated at approximately 1–3% of the population [2], with wide variations in etiology, severity, and associated comorbidities. According to the American Psychiatric Association [3], ID is characterized by significant limitations in both intellectual functioning (e.g., learning, reasoning, problem solving) and adaptive behavior, which includes a broad range of social and practical skills used in everyday life. These deficits typically emerge during the developmental period and affect multiple cognitive domains. The assessment of intellectual functioning emphasizes skills across conceptual, practical, and social areas, such as self-care, social participation, and academic performance [4].

The causes of ID are multifactorial and may arise during the prenatal, perinatal, or postnatal stages. Prenatal causes include genetic syndromes, metabolic conditions, cerebral malformations, and maternal illnesses. Perinatal complications, such as birth asphyxia leading to neonatal encephalopathy and postnatal factors—like hypoxic–ischemic injury, traumatic brain injury, infection, epilepsy, and metabolic disorders—further contribute to its etiology [3,5,6]. Among the known causes, genetic conditions are the most frequent. Down syndrome is the most common chromosomal abnormality, while fragile X syndrome is the leading inherited cause of ID [7,8]. ID often co-occurs with other neurodevelopmental and neuropsychiatric conditions, such as autism spectrum disorder and epilepsy. It may also be associated with neuromuscular impairments, including ataxia, spasticity, neuropathies, or muscular dystrophies [5,9,10].

According to the DSM-5 [3], the severity of ID ranges from mild to profound and is based on the level of adaptive functioning rather than IQ alone. Children with ID typically require ongoing support with daily tasks, with the type and intensity of care depending on the degree of impairment [11]. These children often remain dependent throughout their lives [12], placing a sustained caregiving burden on families. In most cases, this role is assumed by parents and extends across the lifespan, affecting many dimensions of family life [13,14,15].

To contextualize this caregiving experience, the introduction will address four key thematic areas that frame the rationale for this study: (1) the impact of caregiving on family quality of life; (2) the development and conceptualization of self-care (SC) and family self-care (FSC); (3) systemic barriers that affect the implementation of FSC in families of children with ID; (4) this study’s aims and relevance.

### 1.1. Impact of Caregiving on Family Quality of Life

Considering that children with ID are predominantly cared for within the family environment [16], despite some additional support from healthcare and education professionals [17], caregiving presents significant challenges. It can affect the family’s financial stability, social life, physical health, and emotional well-being [18,19]. Several studies have highlighted the lack of structured support for families of children with ID. For instance, the meta-analysis by Dunst, Trivette, and Hamby [17] found that relational and participatory family-centered practices were consistently associated with improved parental satisfaction, family functioning, and child outcomes. However, while the impact of caregiving stress is well documented, less is known about how families actively engage in strategies to sustain their own well-being.

### 1.2. Development of Self-Care (SC) and Family Self-Care (FSC)

Children’s daily care needs—such as assistance with feeding, hygiene, mobility, and communication—require considerable time, coordination, and sustained effort [20]. These responsibilities often disrupt family functioning and contribute to elevated stress, anxiety, and diminished caregiver well-being [21,22]. Additionally, navigating healthcare systems, coordinating specialized services, and balancing professional and family obligations further challenge families’ ability to engage proactively in care and SC activities [21]. A systematic review by Mestre et al. [21] demonstrated that while family-centered care (FCC) approaches can improve caregiver satisfaction, emotional well-being, and service accessibility, their implementation remains inconsistent and context-dependent.

In this study, SC refers to individual practices aimed at maintaining physical, emotional, and social well-being. FSC, by contrast, encompasses how these practices are collectively understood, negotiated, and enacted within the family unit, particularly in caregiving contexts. While SC includes personal strategies such as rest, recreation, or seeking support, FSC emphasizes shared responsibilities, relational interdependence, and the systemic conditions that enable or constrain care within the family.

FSC is conceptually grounded in Orem’s Self-Care Theory [23], Riegel’s Middle-Range Theory of Self-Care in Chronic Illness [24], von Bertalanffy’s General Systems Theory [25], and Casey’s Care Partnership Model [26]. It views the family not merely as a caregiving unit but as a dynamic system in which health behaviors are co-constructed and sustained over time [27]. This relational approach is particularly relevant for families of children with ID, who often require long-term, non-delegable care. The World Health Organization [28] defines SC as the capacity of individuals, families, and communities to promote health and well-being—with or without the support of healthcare professionals.

The recent literature advocates for FSC as a critical care pattern for families of children unlikely to achieve SC autonomy due to the severity of their conditions [22,27]. Despite its relevance, however, FSC remains an underdeveloped concept in both research and practice. Most healthcare interventions still focus primarily on individual patients, often overlooking the family as an integrated agent of care.

The emerging FSC framework [27] identifies four interrelated domains:Cognitive: encompassing knowledge, beliefs, and health literacy;Behavioral: involving routines, help-seeking, and proactive care strategies;Physical: including access to services, environmental adaptations, and caregiving skills;Psychosocial: covering emotional resilience, cohesion, coping, and mental well-being.

These domains are shaped by both internal family dynamics and external factors such as the access to healthcare and the availability of social support [22,27].

FSC is thus understood as a dynamic, multidimensional care pattern involving these four competencies, enabling families to care for themselves and their vulnerable members [22,27]. Existing studies suggest that effective FSC improves the quality of life for the entire family unit [27,29]. Nevertheless, there is limited research on how families actually develop and sustain these practices in everyday life—particularly in under-resourced settings.

This study addresses that gap by qualitatively exploring the experiences, strategies, and challenges of families raising children with various types and severities of ID. Through semi-structured interviews, this research investigates how FSC is constructed in real-life caregiving contexts and identifies key factors—such as family structure, caregiving roles, professional relationships, and resource access—that either support or hinder its development.

### 1.3. Barriers to Access and Systemic Support

Several studies have demonstrated that when health professionals adopt an FCC approach—promoting shared decision-making [30,31], recognizing family expertise [30,32], and encouraging active family participation [30,31,32,33]—families report greater satisfaction with services, enhanced caregiving efficacy, and an increased sense of empowerment [30,31,32,33]. FCC models foster collaborative care planning and emphasize the value of family knowledge alongside shared decision-making between professionals and caregivers.

However, the implementation of FCC remains inconsistent across healthcare systems. A systematic review by Mestre et al. [21] synthesized the effects of FCC interventions on families of children with ID, reporting improvements in caregiver satisfaction, emotional well-being, and access to services. Nonetheless, the review also identified persistent barriers such as fragmented care, limited provider training, and a lack of consistent integration of FCC principles into everyday clinical practice. These findings underscore the need for more applied, context-sensitive research to explore how FCC can be effectively embedded in daily caregiving—particularly in under-resourced or geographically isolated settings.

Additional evidence confirms that FCC is often poorly implemented in community-based care, lacking consistent integration into routine care plans [34,35]. Although the literature acknowledges the central role families play in supporting children with ID, limited attention has been given to how families actually construct, develop, and sustain FSC practices in their daily lives.

The few studies that explore the family’s role in SC primarily focus on children with chronic illnesses [24,36,37], offering only limited insight into the specific context of IDs. Despite the increased attention to informal caregiving, gaps remain in understanding the FSC strategies used by families of children with ID. This gap becomes even more pronounced when considering how social determinants of health—such as education level, income, household size, healthcare access, and social support—influence the development of FSC as a care pattern [22].

Previous research has largely concentrated on clinical or therapeutic issues [20,38,39,40], often overlooking the everyday strategies families adopt to cope with the multifaceted challenges of raising a child with ID.

### 1.4. Study Aim and Relevance

By giving a voice to families, this study aligns with FCC frameworks that emphasize the importance of empowering and supporting families as both recipients and providers of care [21]. It also reflects the guidance of international health organizations advocating investment in community-based, family-oriented care models that enhance sustainability and the quality of life for vulnerable populations [41,42].

The main objective of this qualitative study is to understand the factors influencing the development of FSC—across its physical, cognitive, psychosocial, and behavioral domains—in families of children with ID. Specifically, this study aims to: (1) explore how families organize daily caregiving and access healthcare and support services; (2) examine the emotional impact of caregiving and the coping strategies used; (3) investigate families’ knowledge of their child’s condition, including perceived information gaps and learning experiences throughout the care process; (4) analyze the dynamics of family interaction, their relationships with healthcare professionals, and perceived levels of social and community support.

Grounded in a systems-oriented and family-centered perspective, this research contributes to the expanding knowledge base on FSC and advocates for its recognition as a vital care pattern in pediatric chronic care.

The growing interest in advancing family-focused nursing knowledge—particularly regarding the unique characteristics of SC—is driven by shifting epidemiological patterns and evolving family structures. The global prevalence of families caring for children with ID and other chronic conditions remains substantial [43]. Consequently, it is essential for family nurses and healthcare professionals to possess the skills and understanding necessary to support, educate, and partner with families as central social units in developing and sustaining FSC practices [28,44].

In this context, the study seeks to explore how families of children with ID construct and sustain FSC across its four interrelated domains and to identify the contextual, relational, and systemic factors that shape its development in everyday caregiving.

## 2. Materials and Methods

### 2.1. Design

This study employed a descriptive–exploratory qualitative design, guided by an abductive analytical approach that integrates both inductive and deductive reasoning [45]. This methodological choice was well suited for investigating FSC—a multidimensional and context-dependent phenomenon encompassing relational, emotional, and behavioral processes within family systems.

The exploratory component facilitated an open-ended investigation into the lived experiences of families raising children with ID, while the descriptive aim focused on identifying and characterizing how these families construct and sustain FSC across four interrelated domains: physical, cognitive, psychosocial, and behavioral. The abductive approach enabled a reciprocal dialog between empirical findings and theoretical constructs, thereby enhancing this study’s capacity to both inform and refine the existing FSC framework [26].

This design was especially appropriate given the limited body of research on FSC and the need for a flexible yet rigorous method to uncover the subjective and systemic factors shaping caregiving dynamics in family contexts.

This study addresses the growing need to understand how family caregivers of children with ID—viewed as a caregiving unit—define, develop, and navigate FSC practices. This inquiry is particularly relevant in contexts characterized by limited formal support, strong cultural expectations surrounding caregiving, and systemic service gaps.

### 2.2. Setting

This study was conducted in Portugal, specifically in the Lower Alentejo region, within the jurisdiction of the Local Health Unit of Lower Alentejo. In a previous phase of the research, families of children with ID had been identified by family nurses working in Primary Health Care across 13 local health centers. In each center, a designated nurse—typically the head nurse—served as the main point of contact with the research team. That initial study aimed to promote family engagement and establish an early connection between participants and the researchers. It also involved the administration of a demographic survey and several validated assessment instruments.

For this second phase, data collection was carried out in the same community settings. The lead researcher contacted families (parents and/or caregivers) who had previously indicated their willingness to participate. Once interest and consent were confirmed, interview appointments were scheduled via telephone.

All interviews were conducted face-to-face, in locations chosen by the families to ensure comfort and convenience—typically in their homes or workplaces. This flexible arrangement was designed to respect each family’s context and foster an environment conducive to open and honest dialog. All interviews were carried out by the lead researcher (TDM).

### 2.3. Participants and Inclusion/Exclusion Criteria

Of the 44 families who participated in the initial quantitative phase of the broader research project, 9 were recruited for this qualitative phase. A non-probabilistic, convenience sampling strategy was used to ensure heterogeneity in caregiving experiences, family composition, and child diagnoses. Recruitment was facilitated through local primary healthcare services, with support from community nurses who identified eligible families and initiated contact. The research team then followed up to explain the study, confirm interest, and arrange interviews.

Initial recruitment included six families, with three additional families added until data saturation was reached [45]. All participants were parents of children with ID. In seven interviews, both mother and father were present; in two cases, only the mother participated. Mothers generally provided more detailed narratives, while fathers offered complementary input. All participants had one child with a confirmed diagnosis of ID.

The inclusion criteria were as follows: participants had to (1) be 18 years or older; (2) serve as the primary caregiver of a child with a formal diagnosis of ID; (3) reside with and care for the child in a home setting; (4) be cognitively capable of participating in an interview; (5) be able to provide informed consent. Additionally, the child had to be over two years of age, with a diagnosis confirmed by qualified health professionals.

Families were excluded if the child was fully institutionalized or if the participant was unable to take part due to language barriers, cognitive limitations, or significant mental health concerns.

All participants gave written informed consent after receiving a clear explanation of the study’s objectives, confidentiality protections, and their right to withdraw at any time.

### 2.4. Interview Guide

To ensure alignment between the interview guide and the study’s objectives, the lead researcher (TDM) conducted two pilot interviews. These preliminary interviews assessed the clarity, coherence, and relevance of the questions. Based on insights gained during this phase, the interview guide was revised and subsequently approved by all authors. The pilot interviews were excluded from the final dataset.

The final interview script was organized into five thematic sections, each designed to explore the core dimensions of FSC. The first section addressed the impact of the child’s intellectual disability on family life, including emotional well-being, daily routines, and interpersonal relationships. The second section focused on both the internal and external factors influencing the family’s capacity for self-care.

The third section investigated how FSC practices were expressed across the four domains of the FSC framework—cognitive, physical, psychosocial, and behavioral—prompting participants to reflect on relevant activities and challenges. The fourth section explored the family’s adaptation to caregiving over time, including coping mechanisms, role adjustments, and efforts to maintain resilience. The fifth and final section invited families to identify support resources within the household and the broader community.

### 2.5. Interviews and Data Collection Procedures

Data collection took place between June and November 2024. The first author (TDM) conducted nine face-to-face interviews, primarily in participants’ homes. Each interview lasted approximately 45 min and was audio-recorded for transcription and analysis. All recordings were transcribed verbatim, and transcripts were reviewed by the first author after re-listening to the audio files. Two additional researchers (APC and EVC) validated the transcriptions. Once the accuracy of the transcripts was confirmed, the audio files were deleted. Field notes and memos were also documented to capture contextual observations and to identify areas for further exploration.

The data collection process involved semi-structured interviews guided by a pre-established script containing both closed and open-ended questions. This flexible format encouraged participants to expand on their experiences, often sharing rich and detailed narratives beyond the initial prompts. Follow-up questions—such as “why” or “how”—were used to probe deeper into relevant themes that emerged organically during the interviews [46]. These interviews were designed to elicit in-depth accounts of FSC practices.

The study followed the Consolidated Criteria for Reporting Qualitative Research (COREQ) guidelines [47], ensuring transparency and methodological rigor. Table 1 outlines the strategies employed to enhance credibility, transferability, dependability, and confirmability in accordance with qualitative research standards.

### 2.6. Ethics

Conducting research with families of children with ID involves significant ethical considerations, particularly due to the emotional sensitivity of the topic and the potential social vulnerability of participants. Although caregivers are not always formally classified as a vulnerable population, their experiences often include cumulative stress, grief, and systemic marginalization [18,19,20,48].

In this study, all interviews were conducted in private settings, enabling the participants to share their experiences at their own pace. Information regarding available psychosocial support services was offered, though no participant requested follow-up assistance.

Particular attention was paid to addressing power dynamics. The participants were clearly informed that their involvement was voluntary and that they could withdraw or decline to answer any question without penalty. Importantly, no member of the research team held a clinical or service-related role in relation to the participants. While community nurses assisted in facilitating initial contact with families, they were not involved in conducting interviews or analyzing data.

All participants provided written informed consent. Their data were anonymized to protect confidentiality. To ensure informed participation, each family received a written document outlining the study’s objectives, procedures, and their rights. These measures were implemented to minimize undue influence and to ensure that participation was based on genuine, informed consent, in accordance with best practices in qualitative research with caregiving populations [49,50].

This study received ethical approval from the Ethics Committee of the University of Évora (reference number UE_22083) on 16 November 2022, and from the Ethics Committee of the Local Health Unit of Lower Alentejo (reference number ULSBA/01/2023) on 13 January 2023, in Portugal. The approvals were granted in accordance with the International Ethical Guidelines for Health-related Research Involving Humans.

### 2.7. Data Analysis

For this study, content analysis was employed as the primary method of data analysis, following Bardin’s thematic–categorical approach [45]. The coding process was guided by inductive reasoning grounded in the participants’ narratives, allowing categories to emerge organically from the data. These emergent categories were subsequently interpreted through a deductive lens, informed by the four domains of FSC. This dual strategy enabled the identification of core meanings within participants’ accounts while anchoring the findings in a well-established theoretical framework.

The analysis followed a structured process comprising: (1) defining the study objectives and theoretical framework; (2) establishing the analytical corpus; (3) formulating categories; (4) identifying units of analysis.

The research objectives and theoretical underpinnings shaped both data collection and interpretation. The analytical corpus consisted of complete transcripts from nine interviews, aligned with Bardin’s methodological standards. Categories—defined as conceptual groupings of similar content—were predominantly developed through inductive reasoning. While content analysis allows for both priori and emergent categories, most categories in this study were derived directly from participants’ discourse.

Each category was defined through a gradual process of analogical classification of recording units using an open, inductive approach. Simultaneously, the FSC framework served as a deductive guide to structure the interpretation. The four domains of FSC provided a conceptual scaffold for organizing and interpreting the inductively derived categories. This blended analytical strategy reflects abductive reasoning—balancing openness to new insights with theory-informed analysis.

To strengthen analytical rigor, three researchers (TDM, MJL, and EVC) independently coded the full transcripts. Each conducted a separate thematic analysis before participating in joint sessions to compare interpretations, resolve discrepancies, and reach a consensus on the final coding framework. Although no inter-rater reliability coefficients (e.g., Fleiss’ Kappa or ICC) were calculated, this process reinforced credibility through investigator triangulation and peer debriefing. NVivo 14 software was used to support data management and maintain consistency in coding.

Two types of analytical units were defined: recording units and context units. A recording unit refers to the smallest segment of content that conveys meaning, while a context unit encompasses the broader segment required for its interpretation [45]. In this study, each participant’s response to a specific question was treated as a context unit.

The analysis began with a preliminary “floating” reading of all transcripts to ensure alignment with the study’s objectives. Transcripts were then read repeatedly to identify recurring themes, contrasts, and salient details. Key excerpts were highlighted and grouped into thematic nodes. Although the analysis was conducted manually, NVivo supported the organization and retrieval of qualitative data [51,52].

The research team engaged in iterative discussions to synthesize the findings, leading to the development of interpretive conclusions that described relational dynamics and patterns within and across the four FSC domains. These conclusions were not pre-established but rather emerged through analytical abstraction, consistent with the principles of naturalistic inquiry [53] and qualitative theory development [54].

In addition to thematic coding, the analysis sought to uncover the deeper meanings embedded in families’ caregiving decisions and behaviors. As Bruner [55] emphasizes, narrative is a fundamental mode through which individuals make sense of experiences—connecting internal emotions with external actions. In his words: “we organize our experience and our memory of human happenings mainly in the form of narrative—stories, excuses, myths, reasons for doing and not doing, and so on” [55].

To ensure rigor and trustworthiness, several strategies were applied throughout the analysis [56]. Triangulation was achieved through collaborative coding and interpretation by three researchers (TDM, MJL, and EVC). A clear distinction was maintained between the participants’ voices and the researchers’ interpretations. Reflexivity was promoted by the fact that none of the researchers had prior relationships with the participants. Member checking was also conducted: transcripts were returned to participants for verification, enhancing the credibility and contextual accuracy [57]. Multiple illustrative quotations were included to support transparency and allow readers to assess the validity of the interpretations [58]. Theoretical saturation was considered achieved when no new categories emerged and all four FSC domains were fully developed and confirmed through consensus within the research team.

### 2.8. Research Team and Reflexivity

The interviews were conducted by the first author, who, at the time of this study, had limited prior experience with qualitative research. To enhance methodological rigor, the data analysis was carried out collaboratively with another member of the research team (EVC), who possesses extensive expertise in qualitative methodologies. Throughout this study, the research team engaged in ongoing reflexive discussions to critically examine personal assumptions, maintain interpretive neutrality, and ensure that the analysis remained firmly grounded in participants’ narratives.

## 3. Results

This study included nine families, comprising a total of 16 participants. In seven interviews, both parents participated, while in the remaining two, only the mothers were interviewed. The participants ranged in age from 29 to 53 years, with a mean age of 40.2 years (SD = 7.17). Most were married or cohabiting (88.95%), with one participant identifying as divorced (11.05%). Educational levels varied—6.25% (n = 1) had completed only primary education; 50% (n = 8) had lower or upper secondary education; and 43.75% (n = 7) held tertiary qualifications—reflecting a range of socioeconomic and cultural backgrounds. The mothers were primarily engaged in informal caregiving, domestic tasks, or administrative work, whereas the fathers were mostly employed in agriculture or service-related occupations.

Monthly household incomes ranged from ≤EUR 600 to over EUR 2000, with the majority (55.6%) reporting incomes between EUR 600 and EUR 1000. For context, Portugal’s national minimum wage in 2024 was EUR 820 per month, as established by Decree-Law 107/2023 [59], indicating that many participating families were living at or near the poverty line. Household sizes ranged from three to five members. These financial constraints, along with caregiving-related expenses—such as transportation, medical equipment, and therapeutic services—exerted significant pressure on family resources and limited their ability to maintain consistent SC routines.

The children’s ages ranged from 4 to 15 years (mean = 8.2 years; SD = 3.67). Diagnoses included cerebral palsy (n = 3; 33.3%), autism spectrum disorder (n = 2; 22.2%), and one case each of microcephaly, Down syndrome, chromosome 2 inversion, and metabolic syndrome (n = 1 each; 11.1%). This represents a heterogeneous sample of neurodevelopmental and genetic conditions associated with ID. The families resided across urban, semi-urban, and remote rural areas, providing a varied sociodemographic context for analyzing FSC dynamics (see Table 2).

Rather than presenting the findings chronologically or following a question-by-question format, the results are thematically organized to offer a comprehensive understanding of how families construct and engage in SC in their everyday lives. Anchored in the conceptual framework of FSC [22,26], the narratives were interpreted across four interrelated domains: (1) physical, (2) cognitive, (3) psychosocial, (4) behavioral. This structure enables a focused exploration of both the challenges and adaptive strategies described by the participants. To preserve the authenticity of these experiences, selected quotations are included to illustrate the key themes within each domain.

Drawing from an inductive content analysis, the thematic categories presented below emerged directly from the participants’ narratives and were subsequently organized according to the four FSC domains. These categories reflect the most salient patterns in the data and offer a conceptual map for understanding how families experience, manage, and adapt to caregiving demands. Table 3 summarizes the categories identified within each domain along with illustrative coded quotations, serving as a foundation for the detailed analysis that follows.

In total, 18 thematic categories were identified: five in the physical domain, four in the cognitive domain, four in the psychosocial domain, and five in the behavioral domain. These categories capture the core dimensions of daily FSC as described by the families and support the subsequent in-depth analysis.

### 3.1. Physical Domain: Dependency, Adaptation, and Access

The physical domain applies to healthcare access, home adaptations, and support for basic activities of daily living (BADLs). It reflects the family’s functional capacity to provide physical care and addresses the practical demands of caring for a child with ID. The analysis of the narratives revealed five interrelated categories within this domain: (1) high dependency in BADLs; (2) structural adaptations and the need for assistive equipment; (3) support networks—both internal (family) and external; (4) access to therapies and healthcare services; (5) geographic and financial barriers to care.

#### 3.1.1. High Dependency in BADLs

All families described their children as being significantly dependent on caregivers for fundamental activities of daily living, including eating, bathing, dressing, toileting, and mobility. In most cases, the level of dependency extended to near-constant supervision, especially in cases where the child had comorbid physical conditions or profound developmental delays.


*“We do everything ourselves. All the daily activities, everything, everything.”*
(F1, Father)


*“He does it, but we have to tell him absolutely everything, as if it were an order. He doesn’t do anything by himself, on his own initiative.”*
(F3, Mother)

The caregiving load was described as physically and emotionally taxing. Many parents reported sleep disturbances and fatigue of their own due to the physical effort required to support the child’s care.


*“He can’t do anything on his own, and he’s heavy. Lifting him for everything wears us out.”*
(F1, Mother)

#### 3.1.2. Structural Adaptations and Equipment Need

To assist their children’s physical limitations and support safe and efficient caregiving, the families implemented a variety of environmental adaptations and integrated assistive technologies into their daily routines. Modifications to the home environment included remodeling bathrooms for accessible bathing, widening doorways, installing ramps, and adapting furniture to fit mobility needs.


*“We removed the stairs and changed the bathroom layout. We even changed the bathtub for easier access.”*
(F4, Mother)

The families also relied on assistive devices such as orthopedic beds, adaptive chairs, bath seats, and mobility aids. These were described as essential tools for maintaining the child’s comfort and reducing the physical strain on caregivers.


*“[…] we’ve adapted the house to make it easier to move around, eliminating some architectural barriers, and we’ve invested in support equipment, such as the articulated bed, to make care more practical.”*
(F2, Mother)


*“She has a bath chair and a special bed. It’s much easier now than when she was smaller.”*
(F9, Mother)

In some cases, the families designed or modified tools themselves, while others purchased items with personal funds when public assistance was unavailable.


*“We had to invest a lot ourselves. There was no public help when we needed it.”*
(F1, Mother)


*“We use a visual routines board and reorganized the house to minimize distractions and make daily tasks easier for him.”*
(F8, Mother)

Despite recognizing their importance, access to such devices and adaptations was often constrained by financial limitations and bureaucratic barriers.


*“She has an electric chair with a headrest that social security gave us […]. The rest we got from the medical board. And then the material she needs is approved, but it takes a while.”*
(F2, Mother)

#### 3.1.3. Support Networks: Internal (Family) and External

Support networks, both internal (within the family) and external (community and institutional resources), emerged as a critical aspect of physical caregiving among the families in this study. These networks offered instrumental assistance with daily care routines, facilitated the coordination of medical and therapy appointments, and contributed to the management of broader household responsibilities. Internal support primarily involved extended family members—especially grandparents—who often played a vital caregiving role. Their involvement frequently enabled primary caregivers to take necessary breaks or attend to other family obligations.


*“My mother is my biggest support […] She helps bathe him and sometimes stays with him so I can go out.”*
(F8, Mother)

Siblings were also integrated into care routines, especially in families with limited financial means or restricted access to formal services. In many cases, this involvement was framed not only as practical support but also as part of the family’s long-term care planning.


*“Her sister already knows how to feed her and change her… she’ll take care of her in the future.”*
(F2, Mother)

External networks, including friends, neighbors, and local associations, also played an important role. These sources provided occasional help with transportation, assisted in home modifications, or organized community-based fundraising efforts.


*“Financially […] we have the help of the community, because if the community doesn’t help, he won’t have access to the therapies and treatments he needs.”*
(F1, Mother)


*“Our friends organized a fundraiser to help us buy him wheelchair.”*
(F4, Mother)


*“The cerebral palsy center is our great support for everything else.”*
(F2, Mother)

However, the presence and effectiveness of external support varied. Some families benefited from community solidarity and responsive local services, while others felt isolated or unsupported, particularly in rural or deprived regions.


*“Where we live, we don’t have much help… it’s just us, all the time.”*
(F5, Father)

This category underscores how caregiving is not carried out in isolation but is embedded within a broader social and familial context. The presence or absence of reliable support networks significantly influenced the sustainability and quality of physical caregiving.

#### 3.1.4. Access to Therapies and Healthcare Services

Access to ongoing therapies—including physiotherapy, occupational therapy, and speech therapy—was regarded by all participants as essential. These services were viewed not only as critical to the child’s development but also as integral to the family’s overall SC capacity, as they helped mitigate the long-term burden associated with untreated health conditions.


*“We still do therapy at home. It helps a lot, and we can’t afford to stop.”*
(F1, Mother)

Some families described how limited health services in their area of residence reduced their opportunities for respite, psychological support, or recreational activities. Public systems often lacked the capacity to provide consistent support, leading many families to rely on private services, highlighting disparities in access.


*“At the health center, appointments are rare and irregular. That’s why we pay for private sessions.”*
(F7, Mother)

The families also highlighted how limited local resources and economic constraints reduced their ability to access formal services or participate in FSC-enhancing activities. In places with fewer social workers and overloaded health services, the families reported fragmented or inconsistent support. These systemic shortcomings disproportionately affected theose with lower incomes or limited transportation options.

#### 3.1.5. Geographic and Financial Barriers to Care

A significant number of families highlighted the challenges posed by geographic isolation and financial hardship. Families living in rural areas often face long travel times to access specialized services, which became particularly difficult to manage without financial resources or flexible work arrangements.


*“We’re far from everything. Every appointment is a major trip, and sometimes we just can’t make it.”*
(F6, Mother)

The cumulative cost of transportation, therapy, equipment, and home modifications created a substantial financial burden. As a result, the families developed alternative strategies, such as informal fundraising, social support networks, and community initiatives.


*“We collect bottle caps and organize events to help cover the costs of therapy and equipment.”*
(F1, Mother)

### 3.2. Cognitive Domain: Knowledge Development, Learning Gaps, and Training Needs

The cognitive domain of FSC refers to how families understand, process, and apply knowledge related to their child’s condition and the demands of caregiving. Four key categories emerged within this domain: (1) knowledge about health condition and specific care needs; (2) gaps in professional guidance; (3) informal learning through lived experience; (4) the need for family-centered training.

#### 3.2.1. Knowledge of Health Condition and Specific Care

The participants displayed varying levels of understanding concerning their child’s diagnosis, related health needs, and appropriate care strategies. In many cases, this knowledge was developed gradually through daily caregiving experiences, rather than being proactively provided by healthcare professionals.


*“I learnt, I learnt, the three of us learnt how to look after M. and we have to have courage for everything.”*
(F2, Mother)


*“It was only after years of living with it that we truly began to understand what he has.”*
(F3, Mother)

Several parents described a functional understanding of their child’s needs—how to feed, position, stimulate, or monitor for signs of distress—despite lacking a formal clinical explanation.


*“We know him better than anyone… every sign, every gesture. We learned what to do by being with him every day.”*
(F4, Mother)

#### 3.2.2. Gaps in Professional Guidance

A recurring theme in the interviews was the absence of clear, consistent information from health professionals. Many families expressed frustration at the lack of practical advice and individualized guidance, particularly at the time of diagnosis and during transitions in care.


*“There isn’t enough information to help parents or families on how to be carers for a child like ours […] it’s us parents who have to struggle with everything.”*
(F1, Mother)


*“No one really explained what it meant. We were left to figure it out alone.”*
(F8, Mother)

Caregivers described missed opportunities for education in clinical settings, where information was either too technical, insufficiently contextualized, or entirely absent.


*“Everything is difficult, and we don’t even want to remember it at first. I needed a lot of support, people with knowledge, but we didn’t have that.”*
(F4, Mother)

#### 3.2.3. Informal Learning Through Experience

In the absence of formal guidance, families turned to informal learning strategies, gaining knowledge through daily routines, trial and error, peer support, and digital platforms such as online forums and social media groups.


*“My therapy was my neighbor who has a son like mine. She helped me and still helps me a lot.”*
(F4, Mother)


*“I learned a lot by observing and by asking other parents in the same situation.”*
(F8, Mother)

This self-directed learning, while resourceful, often resulted in uneven knowledge and emotional strain, especially when families were unsure whether their practices aligned with clinical standards.


*“We do what we think is right, but sometimes I wonder if it’s enough or if we’re doing something wrong.”*
(F2, Mother)

#### 3.2.4. Demand for Family-Centered Training

The families expressed a strong desire for accessible, hands-on training tailored to their child’s specific needs. The topics mentioned included feeding via percutaneous endoscopic gastrostomy, positioning, communication techniques, and articular mobility exercises.


*“We need practical guidance […] how to lift her, how to help her communicate.”*
(F2, Mother)


*“I think more training is needed […] especially PEG feeding and hygiene care […] and also how to communicate.”*
(F1, Mother)

The participants emphasized that such training should be integrated into the care process, provided at appropriate stages, and responsive to the realities of family life.


*“If they put a bit more heart into what they do, and tried to look at both sides, also look at the side of the patient’s family, it would be easier to understand and help us.”*
(F3, Mother)


*“We don’t need lessons […] we need tools we can use every day at home.”*
(F7, Father)

This domain highlights how families engage cognitively as informal caregivers, underscoring the need for health systems to support them as care partners with timely and practical information.

### 3.3. Psychosocial Domain: Emotional Burden, Adaptation, and Social Perception

The psychosocial domain of FSC encompasses the emotional, relational, and social aspects of caregiving. It reflects how families cope emotionally with their child’s condition, seek social support, and interpret their experiences within a broader societal context. Four interrelated categories emerged: (1) stress and emotional overload, (2) the emotional impact of caregiving, (3) support networks and resilience, (4) stigma and social exclusion.

#### 3.3.1. Stress and Emotional Overload

The caregivers reported high levels of stress, often linked to the constant demands of care, sleep deprivation, and the lack of time for themselves. Many expressed a sense of being physically and emotionally exhausted, especially when managing caregiving with employment or other responsibilities.


*“It’s exhausting. There’s never a break—not even during the night.”*
(F4, Mother)


*“I get no rest, no time for myself. Some days I feel like I’m running on empty.”*
(F8, Mother)

Stress was frequently described as cumulative, intensifying with the child’s level of dependency and the unpredictability of daily routines.

#### 3.3.2. Emotional Impact of Caring

Beyond stress, the caregivers described deep emotional responses tied to their caregiving role, including fear, sadness, guilt, and anxiety, particularly concerning the future. Some described a grieving process for the life they had expected or imagined for their child.


*“It was hard at the beginning… I felt like I lost the life I had planned for him.”*
(F4, Mother)

Despite these emotions, many caregivers also reported moments of joy, fulfillment, and pride in their child’s progress and personality, underscoring the complexity of their emotional experience.


*“Even with all the difficulties, she brings us so much love. It’s not all pain.”*
(F2, Father)

#### 3.3.3. Emotional Support Network and Resilience

Support from partners, extended family, and close friends was crucial in managing emotional demands. The caregivers who had strong emotional support described feeling more capable and less isolated in their role.


*“When couples or families aren’t united, things don’t go well.”*
(F1, Mother)


*“My husband and I are a team. That makes everything more bearable.”*
(F4, Mother)

Resilience was also shaped by internal factors such as hope, spiritual beliefs, and reframing challenges as growth opportunities.


*“We’ve learned to appreciate small things. This journey changed our whole perspective.”*
(F9, Mother)

#### 3.3.4. Stigma and Social Exclusion

The caregivers frequently experienced social stigma, often reporting discomfort in public spaces, a lack of understanding from others, or outright exclusion from services and community life.


*“People look at us with pity or judgment. It’s easier to stay home.”*
(F6, Mother)

Such experiences were combined by a perceived lack of institutional readiness to accommodate children with disabilities, reinforcing feelings of marginalization and invisibility.


*“Schools and public places […] they are not made for our kids. We always have to adapt.”*
(F9, Mother)

This domain reflects the complex emotional landscape that families navigate. While caregiving often leads to significant psychological strain, it is also marked by moments of connection, personal growth, and resilience drawn from meaningful familial and social bonds. Nonetheless, persistent stigma and social exclusion continue to pose critical barriers to emotional well-being.

### 3.4. Behavioral Domain: Routines, Strategies, and Interaction with Care Systems

The behavioral domain of FSC encompasses the concrete actions, routines, and practical strategies that families implement to manage the daily demands of caregiving. The analysis identified five key categories within this domain: (1) the organization and management of care; (2) family well-being routines; (3) coping and resilience strategies; (4) long-term planning for continuity of care; (5) interaction and relationships with health professionals.

#### 3.4.1. Organization and Management of Care

The families developed highly structured routines to navigate the intensive caregiving demands. These routines included fixed schedules for meals, therapies, hygiene, and sleep, which helped maintain stability and reduce stress.


*“We have everything planned—the time for her bath, therapy, meals. Without it, we would be lost.”*
(F9, Mother)

Care responsibilities were often shared between parents, with siblings and grandparents contributing to many households. This division of labor was essential for avoiding caregiver burnout.


*“We usually share it between mum and dad […] during the day the routine and the activities that are done are always divided between me and the father, so that we don’t overload either one […].”*
(F1, Mother)

#### 3.4.2. Family Well-Being Routines

In parallel with care routines, the families also described efforts to maintain their own well-being. These included making time for leisure activities, ensuring emotional bonding with the child through play, and preserving family rituals that promote cohesion.


*“Play is essential. Even if it’s just 20 min, it’s our way of connecting with her and unwinding.”*
(F7, Mother)

Small but meaningful habits, such as shared meals, walks, or watching television together, helped reduce tension and supported the emotional health of all family members.

#### 3.4.3. Coping and Resilience Strategies

The families reported using a range of coping strategies to manage stress and sustain caregiving over time. These included emotional reframing (focusing on the child’s abilities rather than limitations), drawing on faith, and participating in parent networks.


*“We try to see the good. She’s taught us so much.”*
(F5, Mother)

Humor, peer support, and setting short-term goals were also reported as resilience practices that helped families remain focused and optimistic in the face of long-term challenges.

#### 3.4.4. Long-Term Planning for Continuity of Care

Many families were already planning for the future, particularly regarding who would assume care responsibilities when the parents could no longer do so. In most cases, siblings were considered future caregivers, and families actively involved them in current routines as a preparatory measure.


*“Her sister helps with everything and knows what to do. We’re teaching her now for the future.”*
(F2, Mother)

This long-term thinking was often accompanied by a desire for formal resources or guidance on succession planning and legal arrangements.

#### 3.4.5. Interaction and Relationships with Health Professionals

The quality of interactions with health professionals significantly influenced the family’s ability to implement and sustain SC behaviors. Positive relationships, characterized by empathy, availability, and collaborative dialog, empowered the families and increased confidence.


*“We have an excellent relationship with the family nurse, she’s friendly and always available.”*
(F6, Mother)

Conversely, the families reported frustration when faced with fragmented care, a lack of follow-up, or professionals who failed to acknowledge their expertise and role in the child’s care.


*“Sometimes they don’t listen. They think they know better, but we’re the ones there every day.”*
(F4, Father)

This domain underscores the proactive, dynamic role families play in caregiving, not only in organizing care but also in sustaining emotional well-being, adapting to long-term demands, and negotiating their role within healthcare systems. These behaviors are both reactive and strategic, and are shaped by the needs of the child and the broader context in which care occurs.

### 3.5. Emergent Conclusions

As an analytical synthesis of the findings across all four domains of FSC, a set of emergent conclusions was drawn from the narratives of participating families. These inductively derived insights reflect core patterns identified in the data and offer a deeper understanding of the systemic, cognitive, emotional, and relational conditions shaping FSC in families of children with ID [45,56]. Although not established a priori, these conclusions represent grounded, evidence-informed interpretations that may inform future research [54] and the development of targeted interventions as follows:The development of FSC is significantly influenced by the availability and quality of both formal and informal social support networks, which are essential to the daily organization and sustainability of care for children with intellectual and developmental disabilities.Families’ knowledge of their child’s condition and related care needs directly affects their ability to adapt, make informed decisions, and develop SC competencies—especially in contexts where professional guidance is limited.Barriers to accessing specialized healthcare, education, and rehabilitation services hinder family autonomy and limit the effectiveness and long-term sustainability of their SC practices.The presence or absence of emotional support plays a critical role in caregivers’ capacity to manage both physical and emotional strain, directly influencing family well-being and the continuity of care over time.

These emergent conclusions reflect a shift from purely descriptive findings to conceptual insights and highlight the complex, multidimensional nature of FSC as it is experienced and enacted by families in real-world caregiving contexts.

## 4. Discussion

This qualitative study is the first to explore the development of family self-care (FSC) as a care pattern among families of children with intellectual disabilities (ID) in Portugal. Its primary objective was to examine the factors influencing FSC through the lens of four theoretical domains previously identified in the literature. This study aimed to investigate how families construct, organize, and sustain FSC practices in daily life.

Existing knowledge in this area has been shaped largely by two conceptual studies that position the family as both a social unit and a caregiving system with evolving self-care (SC) practices [26,28].

Methodologically, this research employed a qualitative design that integrated both inductive and deductive strategies. Thematic categories and emergent conclusions were generated inductively from participants’ lived experiences, while interpretation was guided by the deductive framework of the FSC domains. This abductive reasoning—linking empirical observation with theoretical reflection—strengthens this study’s contribution to both empirical understanding and conceptual advancement.

Through inductive content analysis, patterns of meaning were synthesized into four emergent conclusions. These conclusions are grounded in participants’ accounts and articulate the systemic, cognitive, emotional, and relational conditions that enable or constrain FSC. Rather than proposing formal hypotheses, these conclusions aim to inform future theory development. They also contribute to the construction of a grounded FSC framework specific to the context of ID.

While no formal FSC model currently exists, the findings of this study extend several foundational theories. Drawing from Orem’s Self-Care Theory [23], the data highlight the importance of both individual SC agency and supportive environments. Riegel’s Middle-Range Theory of Self-Care in Chronic Illness [24] informs our emphasis on decision-making and SC maintenance, reframed here as shared and distributed processes within the family unit. Von Bertalanffy’s General Systems Theory [25] provides a lens through which the family is understood as an interdependent system, where caregiving and SC are mutually reinforcing. Finally, Casey’s Care Partnership Model [26] supports the notion of collaboration between health professionals, children, and families—emphasizing mutual trust, negotiated goals, and active participation. Together, these theories provide a preliminary FSC framework that accounts for the relational, structural, and systemic conditions of care.

The findings point to a necessary departure from individualistic or clinically centered models of care. Instead, FSC emerges as a situated and collective process, shaped by a dynamic interplay of emotional, environmental, institutional, and social factors. SC, in this context, is not a set of isolated actions but a negotiated, evolving process embedded in the everyday routines and realities of family life.

All families in this study were in life stages characterized by the care of young children and/or adolescents. At this stage, families focus primarily on ensuring the survival, development, and well-being of their children [13,60]. The participants described deeply engaged caregiving routines that required physical effort, emotional regulation, careful planning, and continuous adaptation. They also emphasized providing love, protection, and emotional support—core functions that reflect their roles as parents and caregivers.

Concerns about the future were frequent, as was the importance placed on maintaining family cohesion. The child’s developmental needs, coupled with broader contextual constraints, shaped each family’s approach to sustaining health and well-being [33,61].

This study was conducted in the Lower Alentejo region of southern Portugal—an area marked by a rich cultural heritage but also significant sociodemographic challenges, including limited infrastructure, regional inequality, and low-income levels [62,63]. Many residents live below the national income average [62,64], and access to healthcare often requires long travel distances [65]. These challenges are not unique to Portugal and echo international caregiving conditions. For instance, Petrillo et al. [66] describes similar systemic burdens on unpaid caregivers in the UK, and the World Health Organization [41] calls for more inclusive global policies to support informal caregivers.

In our study, the participants described frequent difficulties accessing therapies, long waiting periods, and a dependence on private or non-profit services to fill gaps left by the public health system. Financial strain was also prevalent, including transportation costs, medical equipment expenses, and other out-of-pocket needs. Most families reported monthly household incomes of EUR 600–EUR1000, supporting an average of four family members—placing many below Portugal’s minimum wage threshold (EUR 820/month in 2024 [59]).

Geographic location further influenced access to services. Six of the nine families lived in rural or semi-urban areas, where health and social resources are often sparse. These families had to travel long distances to access specialized care, exacerbating both financial and time-related stress. Such findings align with European research that highlights geographic disparities and service distribution inequalities as critical barriers to SC integration [67,68]. Similarly, Liaqat et al. [69] documented how systemic access issues in rural Pakistan contribute to caregiver stress—reinforcing the cross-cultural relevance of these challenges.

Ultimately, our results demonstrate that FSC is not only relational but also structurally contingent. Geographic and financial inequalities directly limit families’ ability to rest, access care, and sustain social support networks [70]. These findings align with international calls to move beyond individual coping models toward systemic, equity-based frameworks that address the contextual realities of caregiving [17,21,41].

Taken together, these structural, geographic, and experiential factors reveal the complexity of sustaining FSC in families living in vulnerable socioeconomic conditions. This study contributes to a deeper, situated understanding of how FSC is shaped through everyday actions, constraints, and adaptive responses. The four interrelated FSC domains—physical, cognitive, psychosocial, and behavioral—offer a comprehensive lens through which to interpret these caregiving experiences, highlighting both the internal family dynamics and broader systemic influences that define FSC in practice.

### 4.1. Physical Domain: Managing Care Through Material and Social Support

In the physical domain, the families described the exhausting nature of daily caregiving tasks—such as feeding, hygiene, mobility, and constant supervision—and the practical adjustments required to meet these demands. These included home modifications, the use of assistive equipment, and the reliance on both formal and informal support networks. This aligns with the first emergent conclusion, which emphasizes the central role of support systems in sustaining caregiving. The involvement of extended family, the availability of accessible local services, and the presence of responsive professionals helped reduce caregiver burden and ensured continuity of care [18,71].

Importantly, the participants in this study often attributed difficulties in accessing services not to personal reluctance, but to limitations and inconsistencies within the services themselves. This contrasts with prior research, which frequently links such barriers to individual attitudes or lack of awareness [72].

Building collaborative and respectful relationships with health professionals also emerged as a key factor in fostering family engagement. The therapeutic alliance is widely recognized as a strong predictor of positive intervention outcomes [73], and strengthening these relationships may help families re-engage with services—particularly after negative or exclusionary experiences [74]. These findings reinforce earlier research underscoring the importance of both relational and material resources in supporting family caregiving efforts [21,75].

### 4.2. Cognitive Domain: Learning to Care in the Absence of Formal Guidance

The families consistently reported that most of their caregiving knowledge was acquired informally—through lived experience, online resources, trial and error, and peer support—rather than through structured guidance from health or social care professionals. This supports the second emergent conclusion: a caregiver’s ability to adapt, make informed decisions, and build confidence is closely linked to the accessibility, reliability, and contextual relevance of the knowledge available to them.

The data revealed a lack of systematic, tailored education for families, particularly during critical moments such as the time of diagnosis or major care transitions. As a result, many parents were compelled to seek information independently. These findings reflect international evidence indicating that many health systems continue to overlook family-centered education, especially in contexts involving chronic and complex care needs [37,76]. Similarly, Guimarães [77] observed that Brazilian families often rely on informal networks and experiential learning to compensate for the absence of professional guidance—echoing our findings within the cognitive domain of FSC.

Research has shown that when information is insufficient or poorly timed, caregiver anxiety increases, decision-making is compromised, and mistrust in services may develop [78,79]. Conversely, access to timely, practical, and individualized information can empower families, enhance engagement, and improve outcomes for both children and caregivers [80,81].

The families in this study exemplify what Borkman [82] termed “experiential experts”—individuals who draw on daily interactions and accumulated insight rather than formal training. While experience-based learning can be highly effective, it may also expose families to misinformation or inconsistent practices, particularly when not supported or validated by professionals [21,83].

These findings highlight the need for participatory, co-designed educational models. Such approaches should blend clinical expertise with caregivers’ lived experiences, ensuring that information is not only provided but also adapted to each family’s specific context and needs [21,84].

### 4.3. Psychosocial Domain: Navigating the Emotional Landscape of Caregiving

In the psychosocial domain, the families expressed a wide range of emotional responses to their caregiving roles. These ranged from chronic stress, fatigue, and anxiety to feelings of satisfaction, meaning, and resilience. The emotional burden was often linked to long-term uncertainty about their child’s future, the cumulative strain of daily care, and frequent experiences of social misunderstanding and marginalization. Many parents described feeling invisible or judged in public spaces and shared the internal weight of anticipatory grief and emotional vigilance.

The findings of this study align with international evidence highlighting the multidimensional burdens and adaptive strategies adopted by families caring for children with ID. For example, a recent study by Egami et al. [85] in Japan emphasized how caregivers’ emotional resilience and problem solving strategies significantly contribute to managing daily routines in the families of children with neurodevelopmental disorders.

These findings support the fourth emergent conclusion: emotional support—whether formal or informal—is essential for sustaining psychological well-being and caregiving capacity over time. The participants who received consistent emotional support from a partner, extended family, or their community reported a greater capacity to regulate stress, reframe challenges, and remain committed to long-term care. This aligns with extensive research showing that the caregivers of children with disabilities face elevated risks of psychological distress, particularly when support systems are weak or absent [86,87]. Emotional resilience in these contexts is not merely an individual trait but is shaped by the access to relational, community, and institutional resources [79,88].

Social stigma also emerged as a significant psychosocial stressor. It reinforced feelings of isolation and limited families’ participation in public and social life. These findings are consistent with the literature on disability studies, which emphasizes how social exclusion and negative attitudes intensify caregiver strain beyond the physical demands of care [89,90].

The importance of peer support and shared experience was another strong theme in participants’ narratives. The families found emotional relief and validation through informal support networks. These connections fostered a sense of social belonging, reduced stress, and supported more adaptive coping strategies [91,92].

Taken together, these insights point to the need for healthcare and social systems to embed structured psychosocial support into FCC. This includes access to counseling, carer respite, and facilitated support groups as essential components of comprehensive caregiving support.

### 4.4. Behavioral Domain: Proactive Strategies and Long-Term Thinking

The behavioral domain of FSC revealed that families act as strategic and proactive agents in the caregiving process. Rather than responding passively to their child’s needs, the participants consistently described forward-looking behaviors, including the establishment of structured daily routines, the division of caregiving tasks, and preparation for future care scenarios.

Many parents discussed involving siblings in caregiving activities. This strategy not only reduced their immediate caregiving burden but also served to build future caregiving capacity—anticipating their own physical decline or eventual absence. Such practices reflect a form of intergenerational care planning, which is increasingly recognized as essential in the context of long-term caregiving for chronic disabilities [93,94].

These findings support the third emergent conclusion: that limited access to specialized services undermines families’ ability to plan for and sustain long-term care. In several cases, the absence of local services, delays in referrals, or poor coordination among providers compelled families to develop their own coping strategies and rely on community-based resources. This aligns with previous research showing that fragmented service systems often force families to “fill the gaps”, assuming roles typically performed by professionals [95,96].

The quality of families’ relationships with health and social service professionals also significantly influenced behavioral responses. The families who felt respected, informed, and involved in decision-making reported greater trust in services and stronger long-term engagement. The importance of relational care and collaborative decision-making echoes the findings from a review by Park et al. [97], which highlighted the positive impact of FCC approaches on caregiver empowerment and satisfaction across various high-income contexts. Collaborative partnerships—marked by open communication and continuity of care—enhanced family autonomy and supported future planning [98,99]. Conversely, families who encountered rigid, dismissive, or inconsistent professionals expressed mistrust, disengagement, and frustration, as similarly reported in other FCC studies [100,101].

Together, these findings highlight the dual importance of family agency and institutional responsiveness in sustaining the behavioral dimensions of SC. They underscore the need for care systems that promote continuity, flexibility, and relational competence—elements that empower families to maintain caregiving capacity while preserving their autonomy and well-being.

### 4.5. Implications for Practice, Policy, and Research

The findings of this study carry significant implications for clinical practice, policymaking, and future research. Although the results are context-specific and not intended for broad generalization, they provide transferable insights that may be applicable in other sociodemographic contexts. The integration of inductive and deductive reasoning—through an abductive, mixed qualitative approach—strengthens the study’s capacity to inform both grounded clinical practice and theoretical advancement.

From a clinical standpoint, the development of FSC among families of children with ID underscores the urgent need to implement genuine FCC models. Health professionals should engage families as active partners in care, acknowledging their experiential knowledge, understanding their daily caregiving routines, and addressing their specific contextual challenges. Supporting FSC requires tailored training, practical and accessible guidance, and collaborative decision-making processes that foster family autonomy and resilience.

The FSC framework—structured around physical, cognitive, psychosocial, and behavioral domains—can serve as a practical guide for developing family education programs. It helps identify gaps in knowledge, emotional support needs, and skill development across the caregiving journey. These programs can be delivered by multidisciplinary teams and adapted to different family structures and caregiving intensities.

In community nursing, the FSC framework can inform protocols that emphasize whole-family assessments, culturally sensitive care planning, and ongoing support tailored to caregivers’ evolving needs. For example, community nursing protocols could incorporate FSC assessments during home visits, allowing nurses to co-develop care plans that reflect families’ capacities, challenges, and goals. These protocols might include periodic check-ins focused not only on the child’s clinical needs but also on caregiver well-being, burden, and available social support. Caregiver training programs could be designed using the FSC framework to deliver targeted modules—such as stress management, assistive care techniques, or navigating local health systems—offered through community health centers or digital platforms. In municipalities, the findings may support the development of respite care initiatives that are flexible, home-based, and tailored to the specific rhythms of family life, especially in under-resourced or rural areas like Lower Alentejo. These programs could be coordinated by local health and social care units, in partnership with family representatives, to ensure that the services align with lived caregiving realities. This care pattern shifts the focus from individual patient care to family-level well-being and long-term caregiving capacity.

At the policy level, this study draws attention to structural inequities—particularly in socioeconomically vulnerable and geographically isolated areas like Lower Alentejo—that block the development of sustainable FSC practices. Policymakers should prioritize equitable access to specialized services, reduce bureaucratic delays (especially regarding assistive equipment and therapies), and invest in both formal and informal community-based support systems. Integrating FSC into disability support policies may enhance the visibility of families as long-term care partners and ensure that systems are designed to support—not replace—their role in sustaining care.

In terms of research, the four emergent conclusions developed through this analysis contribute to building a context-sensitive, multidimensional care pattern of FSC. Future longitudinal studies are needed to explore how FSC evolves over time, particularly as children with ID transition through different life stages. Further research should also examine how intersecting social determinants—such as income, education, and geographic access—influence the development of FSC. Comparative studies across diverse cultural and healthcare systems would help refine the FSC framework and support the creation of targeted, evidence-based interventions.

### 4.6. Strengths and Limitations

One of the key strengths of this study is its mixed qualitative approach, combining inductive and deductive reasoning within an abductive analytical framework. This facilitated the emergence of rich, theory-generating insights grounded in families’ lived experiences, while maintaining alignment with the established family self-care (FSC) care pattern. The use of Bardin’s thematic–categorical content analysis, supported by NVivo 14 software and reinforced through investigator triangulation, enhanced analytical rigor and credibility.

Moreover, this study represents one of the first applications of the FSC framework in the context of families of children with intellectual disability (ID) in Portugal, offering a context-sensitive and empirically grounded contribution to the field.

Nonetheless, some limitations should be acknowledged:

First, the use of a convenience sampling strategy may have introduced selection bias, as families more engaged with community networks or services were more likely to participate. The sample was relatively small (n = 9 families) and geographically confined to the Lower Alentejo region of Portugal. This limits the statistical power, reduces the variability of perspectives, and constrains the transferability of findings to other regions or sociocultural contexts. The local healthcare systems, institutional resources, and cultural understandings of caregiving likely influenced the results.

Second, although both mothers and fathers were present in most interviews, maternal voices predominated. This may be due to differences in verbal expressiveness and could have shaped the representation of caregiving dynamics, particularly in regard to paternal roles.

Third, while a community nurse helped facilitate participant contact, data collection was carried out by researchers unaffiliated with caregiving services. Nonetheless, some participants may have perceived a connection with healthcare institutions, potentially contributing to social desirability bias—where participants present their care practices more favorably. Additionally, this study relied entirely on self-reported narratives, which are inherently subject to recall bias or emotional filtering.

Despite these limitations, this study’s deep engagement with participants and the integration of grounded narratives with theoretical frameworks offers a rich and context-sensitive contribution. The findings provide relevant implications for family nursing, disability research, and public health policy—particularly in promoting inclusive, family-centered care (FCC) models in under-resourced settings.

## 5. Conclusions

This qualitative study offers a comprehensive and context-sensitive understanding of how families of children with ID construct and sustain FSC. Drawing on the narratives of nine families in southern Portugal, this research examined how FSC practices are developed and adapted across four interrelated domains: physical, cognitive, psychosocial, and behavioral.

Families were found to actively manage caregiving through environmental adaptations, experiential learning, emotional resilience, and long-term planning. These strategies were shaped by limited access to services, regional disparities, and the quality of relationships with health professionals—highlighting the urgent need for more inclusive and family-centered support systems.

Importantly, this study shows that FSC is not a static or purely individual activity. Rather, it is a dynamic, relational, and evolving process shaped by daily routines, emotional coping, social support, and structural conditions. Using an abductive analytical approach, this study integrated themes that emerged inductively from the data with a deductively applied theoretical framework. This led to the development of four emergent conclusions that emphasize the importance of social support, health literacy, service accessibility, and emotional well-being in enabling sustainable and effective FSC practices.

By situating families’ experiences within their sociodemographic and geographic realities, this study reveals the structural barriers that limit the development of FSC—and the adaptive strategies families use to overcome them. These findings call for urgent investment in family-centered policies and interventions that recognize caregivers as essential health agents. Such policies should provide families with the knowledge, resources, and emotional support needed to sustain long-term care.

This study contributes to the theoretical advancement of FSC as a multidimensional care pattern for understanding and supporting family caregiving in the context of pediatric ID. It also offers actionable insights for clinicians, educators, and policymakers seeking to improve the quality of life and care outcomes for children with ID and their families.

While the findings offer meaningful insights, they remain exploratory and context-specific. Given the small sample size and the focus on a single geographic region (southern Portugal), the results are not generalizable to all families or care contexts. Instead, they should be viewed as an in-depth, situated contribution to understanding how FSC emerges within a specific sociocultural and healthcare environment.

Future research is essential to build on these findings. Quantitative and mixed-methods studies are particularly needed to validate the FSC framework, assess its applicability across different family structures, and examine its impact on health outcomes and caregiving sustainability. Longitudinal and cross-cultural studies would also help determine how FSC evolves across the life course and in response to diverse structural conditions. Such work will be key to transforming FSC from an emergent conceptual care pattern into a robust, evidence-based tool for practice and policy.

## Figures and Tables

**Table 1 healthcare-13-01705-t001:** Strategies for ensuring qualitative rigor (aligned with COREQ guidelines).

Rigor Criterion	Strategies Applied in This Study
Credibility	- Triangulation among three researchers during coding and category development - Member checking (transcripts returned to participants for verification) - Use of direct quotes to support interpretations
Transferability	- Substantial description of participant context, caregiving roles, and sociogeographic setting - Clarification of inclusion criteria and sampling procedures
Dependability	- Detailed documentation of the analytical process (Bardin’s content analysis approach) - Use of NVivo 14 software for data organization and audit trail
Confirmability	- Reflexivity ensured by lack of prior relationships between researchers and participants - Clear distinction between raw data and interpretation - Consensus-based validation of themes across the research team

**Table 2 healthcare-13-01705-t002:** Sociodemographic characteristics of the sample.

Interviewees	Participant(s)	Age(s)	Marital Status	Educational Level	Occupation	Household Income	House-Hold Size	Type of ID	Age of the Child with ID	Place of Residence
F1	Mother Father	32 35	Cohabiting	Higher Education Higher Education	Agronomist Informal Caregiver	EUR 600–EUR 1000	4	Cerebral Palsy	6	City
F2	Mother Father	49 46	Cohabiting	Primary Education Secondary Education	Informal Caregiver Agricultural Worker	EUR 600–EUR 1000	4	Cerebral Palsy	15	City
F3	Mother	46	Married	Secondary Education	Operational Assistant	>EUR 2000	5	ASD	11	City
F4	Mother Father	40 42	Married	Secondary Education Secondary Education	Cook Waiter	EUR 600–EUR 1000	3	Microcephaly	6	Town
F5	Mother Father	29 31	Cohabiting	Secondary Education Secondary Education	Housekeeper Agricultural Worker	EUR 600–EUR 1000	4	Down Syndrome	6	Village
F6	Mother Father	48 53	Married	Secondary Education Higher Education	Housekeeper Agricultural Worker	≤EUR 600	3	Chromosome 2 inversion	11	Rural Homestead
F7	Mother Father	34 37	Cohabiting	Higher Education Higher Education	Food Engineer Agricultural Engineer	EUR 1500–EUR 2000	4	Metabolic Syndrome	4	Town
F8	Mother	35	Divorced	Secondary Education	Administrative	EUR 600–EUR 1000	3	ASD	5	Village
F9	Mother Father	44 42	Married	Higher Education Higher Education	Psychologist Office Assistant	>EUR 2000	3	Cerebral Palsy	10	Town

Legend: ASD—Autism Spectrum Disorder; F—Family; Cohabiting—unmarried, living with a partner.

**Table 3 healthcare-13-01705-t003:** Categories and coded quotations identified in each domain of family self-care.

FSC Domain	Categories	Coded Quotations
Physical	High dependency in basic activities of daily living (BADLs)	18
	Structural adaptations and equipment need	9
	Support networks: internal (family) and external	14
	Access to therapies and healthcare services	11
	Geographic and financial barriers to care	12
Cognitive	Knowledge of health condition and specific care	15
	Gaps in professional guidance	10
	Informal learning through experience	13
	Demand for family-centered training	8
Psychosocial	Stress and emotional overload	17
	Emotional impact of caring	16
	Emotional support network and resilience	12
	Stigma and social exclusion	9
Behavioral	Organization and management of care	20
	Family well-being routines	14
	Coping and resilience strategies	19
	Long-term planning for continuity of care	11
	Interaction and relationships with health professionals	10

## Data Availability

The datasets used and/or analyzed during the current study are available from the corresponding author upon reasonable request.

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
