# Peer review of "Family Self-Care in the Context of Intellectual Disabilities: Insights from a Qualitative Study in Portugal"

_healthcare, 2025, doi:10.3390/healthcare13141705_

Round 1
Reviewer 1 Report
Comments and Suggestions for Authors
The authors refer to Orem's Self-Care Theory, Riegel's Middle-Range Theory of Self-Care in Chronic Illness, and von Bertalanffy's General Systems Theory but do not provide references to the sources describing them. Footnotes and a bibliography should supplement this regard.
By writing that “Several studies have demonstrated that when health professionals adopt a Family- Centered Care (FCC) approach - promoting shared decision-making, recognizing family knowledge, and encouraging active family participation - families report increased satisfaction with services, improved caregiving efficacy, and a sense of empowerment [21]” the authors refer to only one position, there are no references to specific studies that report on this. These should be included.
In the excerpt (lines 178-179) there is no reference to the article discussing Bardin's content analysis. It would be good to add this.
It would be advisable for authors to describe the research problems in the section devoted to the research method, in addition to clearly defining the research objective.
It is valuable to draw a set of theoretical propositions from the narratives of the mothers and fathers studied. The authors refer to them as "inductively derived hypotheses" (line 649). In qualitative research, hypotheses are usually not formulated in such a precise and formal manner as in quantitative research. In qualitative research, research questions are more often formulated that are exploratory and help understand the phenomenon, rather than verify specific hypotheses. What the authors describe are rather conclusions from the research, not hypotheses.
Author Response
Response to Reviewer 1
Manuscript ID: healthcare-3698252
Title: Family Self-Care in the Context of Intellectual Disabilities: Insights from a Qualitative Study in Portugal
Comments and Suggestions Sent:
1 - The authors refer to Orem's Self-Care Theory, Riegel's Middle-Range Theory of Self-Care in Chronic Illness, and von Bertalanffy's General Systems Theory but do not provide references to the sources describing them. Footnotes and a bibliography should supplement this regard.
Response: We thank the reviewer for highlighting the need to include references for the theoretical frameworks that support our analysis. In response, we have revised the manuscript to explicitly cite the foundational works describing each model referenced in our discussion of the Family Self-Care (FSC) framework (please see references highlighted and numbered from 23 to 26). We also incorporated Casey’s Model of Partnership in Care, which reinforces the relational and collaborative nature of family-centered care. This model supports our emphasis on shared responsibility between families and professionals, and we have integrated its principles into our conceptual framing of FSC. The relevant citation has now been added to the manuscript and reference list.
2 - By writing that “Several studies have demonstrated that when health professionals adopt a Family- Centered Care (FCC) approach - promoting shared decision-making, recognizing family knowledge, and encouraging active family participation - families report increased satisfaction with services, improved caregiving efficacy, and a sense of empowerment [21]” the authors refer to only one position, there are no references to specific studies that report on this. These should be included.
Response: We appreciate the reviewer’s observation regarding the need to reference specific studies that support the reported benefits of Family-Centered Care (FCC). In response, we have revised the sentence to include multiple references to recent empirical studies and systematic reviews that demonstrate the positive outcomes associated with FCC approaches.
3 - In the excerpt (lines 178-179) there is no reference to the article discussing Bardin's content analysis. It would be good to add this.
Response: Thank you for pointing this out. We have now added the appropriate reference to Bardin’s content analysis method to support the mention in the previous lines 178–179. You can see it now, on lines 200-201.
4 - It would be advisable for authors to describe the research problems in the section devoted to the research method, in addition to clearly defining the research objective.
Response: Thank you for this helpful suggestion. In response, we have revised the Methods section to include a brief description of the research problem guiding the study. Specifically, we now clarify that the research addresses the limited understanding of how family caregivers of children with ID (seen as a unit) conceptualize and practice self-care, in the context of constrained resources and limited formal support in Portugal. This complements the clearly stated objective already present in the Introduction. The revised content appears in the opening paragraph of the 2.1 Design section.
5 - It is valuable to draw a set of theoretical propositions from the narratives of the mothers and fathers studied. The authors refer to them as "inductively derived hypotheses" (line 649). In qualitative research, hypotheses are usually not formulated in such a precise and formal manner as in quantitative research. In qualitative research, research questions are more often formulated that are exploratory and help understand the phenomenon, rather than verify specific hypotheses. What the authors describe are rather conclusions from the research, not hypotheses.
Response: Thank you for your thoughtful and accurate observation. We agree that the term “inductively derived hypotheses” may be misleading in the context of qualitative research, where the emphasis is on generating exploratory insights rather than testing formal hypotheses. In response, we have revised the language in the Results, Discussion and Conclusion sections to reflect that these are emergent conclusions drawn from the participants’ narratives, rather than formal hypotheses. We believe this revision better aligns with qualitative methodology.

Reviewer 2 Report
Comments and Suggestions for Authors
This is a timely study that contributes to the growing field of family-centered care by exploring the multidimensional practice of Family Self-Care (FSC) in families with children with intellectual disabilities (ID). The paper is well-structured, methodologically sound, and grounded in both empirical data and theoretical frameworks. It offers significant insights for clinical practice, policy, and future research. However, several areas require refinement or clarification to enhance the paper's rigor, coherence, and impact.
The paper introduces theoretical propositions derived from the data, but their contribution to the evolution of FSC theory remains somewhat abstract. Strengthen the discussion by explicitly comparing these findings with existing models of FSC or related caregiving theories. Clarify how this study extends or modifies current theoretical understandings.
The limitations section is brief and does not fully engage with the methodological and contextual constraints. Expand to address: Potential bias from convenience sampling. Limited generalisability due to small, homogeneous sample (e.g., most participants were mothers). Impact of interviewer–participant relationship (especially as the lead researcher was also a community nurse).
While the influence of geographic and economic constraints is mentioned, their analytical integration across all domains could be stronger. Consider threading the socioeconomic context (especially regional inequities) more consistently throughout the results and discussion sections to highlight systemic structural determinants.
There is minimal reflection on the ethical dimensions of researching a vulnerable population. Discuss any ethical tensions or dilemmas encountered (e.g., power dynamics, emotional burden of interviews) and how these were mitigated to increase the paper’s trustworthiness.
Sometimes “SC” and “FSC” are used interchangeably; clarify early and maintain consistent use throughout.
Theoretical Proposition - the term is used to describe grounded insights, but may mislead readers expecting testable hypotheses. Consider rephrasing or clearly defining the use in qualitative context. Consider - "Interpretive propositions” “Grounded conceptual insights” “Emergent conceptual patterns” “Analytical assertions”
Include more international sources for broader relevance, if possible.
Some sentences are overly long or complex (especially in the methods and discussion sections). Consider simplifying for clarity.
May need to Clarify Socioeconomic Data, household income brackets contextualised for international readers (e.g., relative to national minimum wage).
There are minor typographical issues (e.g., spacing, punctuation in lists) that should be reviewed for publication.
Uses - Consolidated Criteria for Reporting Qualitative Research (COREQ).
Author Response
Response to Reviewer 2
Manuscript ID: healthcare-3698252
Title: Family Self-Care in the Context of Intellectual Disabilities: Insights from a Qualitative Study in Portugal
Comments and Suggestions Sent:
1 - This is a timely study that contributes to the growing field of family-centered care by exploring the multidimensional practice of Family Self-Care (FSC) in families with children with intellectual disabilities (ID). The paper is well-structured, methodologically sound, and grounded in both empirical data and theoretical frameworks. It offers significant insights for clinical practice, policy, and future research. However, several areas require refinement or clarification to enhance the paper's rigor, coherence, and impact.
Response: Thank you very much for your appreciation.
2 - The paper introduces theoretical propositions derived from the data, but their contribution to the evolution of FSC theory remains somewhat abstract. Strengthen the discussion by explicitly comparing these findings with existing models of FSC or related caregiving theories. Clarify how this study extends or modifies current theoretical understandings.
Response: Thank you for highlighting this important point. As there are currently no formal models of Family Self-care (FSC) in the context of ID, our study aims to contribute to theory development by identifying foundational categories from caregivers lived experiences. In the revised Discussion, we explicitly relate our findings to elements of Orem’s Self-Care Theory, Riegel’s Middle-Range Theory of Self-Care in Chronic Illness, von Bertalanffy’s General Systems Theory, and Casey´s Care Partnership Model. This strengthens the conceptual grounding of the proposed framework and clarifies how it builds upon and integrates existing theoretical traditions to address the family caregiving context. Please see the Discussion section – lines 767 to 783.
3 - The limitations section is brief and does not fully engage with the methodological and contextual constraints. Expand to address: Potential bias from convenience sampling. Limited generalisability due to small, homogeneous sample (e.g., most participants were mothers). Impact of interviewer–participant relationship (especially as the lead researcher was also a community nurse).
Response: Thank you for this valuable observation. We have expanded the Limitations section to address potential biases related to convenience sampling, the demographic profile of the sample, and the interviewer–participant relationship. We would also like to clarify that the lead researcher was not a community nurse. A community nurse affiliated with the region facilitated initial contact with some families, but was not involved in data collection or analysis. We have revised the manuscript accordingly to ensure transparency and avoid misinterpretation (Strengths and Limitations section).
4 - While the influence of geographic and economic constraints is mentioned, their analytical integration across all domains could be stronger. Consider threading the socioeconomic context (especially regional inequities) more consistently throughout the results and discussion sections to highlight systemic structural determinants.
Response: Thank you for this important suggestion. We have reviewed the Results and Discussion sections to more explicitly and consistently thread the influence of geographic and socioeconomic inequities across all categories. We now emphasize how limited access to formal services, regional disparities in health and social infrastructure, and economic privation shape SC capacity and expectations among family caregivers. These structural factors are presented not only as background but as embedded constraints that interact with the families lived experiences. The revised passages aimed to strengthen the focus on systemic determinants – Highlighted on the subsection 3.1.4. Access to therapies and healthcare services, and on the Discussion section - lines 810–813 and 827-846.
5 - There is minimal reflection on the ethical dimensions of researching a vulnerable population. Discuss any ethical tensions or dilemmas encountered (e.g., power dynamics, emotional burden of interviews) and how these were mitigated to increase the paper’s trustworthiness.
Response: Thank you for highlighting this important aspect. We agree that greater attention to the ethical dimensions of conducting research with families of children with ID is essential. In response, we have expanded the Methods section to include a reflection on ethical tensions and mitigation strategies. Specifically, we discuss how power dynamics, emotional vulnerability, and the potential burden of sharing sensitive experiences were considered throughout the research design and data collection (On the subsection 2.6. Ethics).
6 - Sometimes “SC” and “FSC” are used interchangeably; clarify early and maintain consistent use throughout.
Response: Thank you for pointing out this important detail. We have revised the manuscript to clearly distinguish between self-care (SC) as an individual practice and family self-care (FSC) as a relational, collective process that occurs within family caregiving contexts. We now define both terms early in the Introduction (lines 93-99) and have reviewed the manuscript to ensure consistent usage throughout.
7 - Theoretical Proposition - the term is used to describe grounded insights, but may mislead readers expecting testable hypotheses. Consider rephrasing or clearly defining the use in qualitative context. Consider - "Interpretive propositions” “Grounded conceptual insights” “Emergent conceptual patterns” “Analytical assertions”.
Response: Thank you for this insightful suggestion. To avoid any confusion with formal, testable hypotheses, we have replaced the term “theoretical propositions” with “emergent conclusions”, which better reflects the interpretive and inductive nature of our qualitative analysis. This phrasing is now used consistently throughout the manuscript and is supported by a clarifying statement in the Discussion section to indicate that these conclusions are grounded in participants’ narratives and derived through interpretive analysis.
8 - Include more international sources for broader relevance, if possible.
Response: Thank you for this helpful suggestion. In response, we have expanded the reference list to include additional international sources - particularly from European, Brazilian, and global health contexts - that address family caregiving, self-care, and systemic support needs. These references provide broader relevance to our findings and situate the Portuguese experience within a wider international discussion on caregiving and family health dynamics. We have integrated these sources into the Introduction and Discussion sections where appropriate.
9 - Some sentences are overly long or complex (especially in the methods and discussion sections). Consider simplifying for clarity.
Response: Thank you for this valuable feedback. We have carefully reviewed the manuscript, particularly the Methods and Discussion sections, and revised several sentences to improve clarity, conciseness, and overall readability. Long or complex sentences were restructured to ensure accessibility for a broad international audience, while preserving the academic tone and meaning of the original content.
10 - May need to Clarify Socioeconomic Data, household income brackets contextualised for international readers (e.g., relative to national minimum wage).
Response: Thank you for this helpful suggestion. We have revised the socioeconomic data to provide clearer context for international readers. Specifically, we now relate the household income brackets to Portugal’s national minimum wage at the time of data collection. This clarification helps readers understand the economic constraints experienced by participating families (please see the Results section (lines 402-409).
11 - There are minor typographical issues (e.g., spacing, punctuation in lists) that should be reviewed for publication.
Response: Thank you for noting the minor typographical issues. We have carefully reviewed the manuscript for spacing, punctuation, and formatting inconsistencies - particularly in lists, hyphenation, and in-text citations. Necessary corrections have been made to ensure clarity and compliance with journal style requirements.
12 - Uses - Consolidated Criteria for Reporting Qualitative Research (COREQ).
Response: Thank you for your observation. We confirm that this study followed the Consolidated Criteria for Reporting Qualitative Research (COREQ) to ensure transparency and rigor in reporting. Although we referenced the COREQ checklist, we acknowledge that its use was not made explicit in the manuscript. We have now incorporated a brief table summarizing the techniques used to ensure qualitative rigor across the four key dimensions: credibility, transferability, dependability, and confirmability (Table 1). This addition aligns with the COREQ guidelines and reinforces the transparency and trustworthiness of our study.

Reviewer 3 Report
Comments and Suggestions for Authors
Thank you very much for the opportunity to participate in the review of this work. It was a great pleasure for me to read and comment on this manuscript. Overall, I would like to congratulate the authors on the structure, theoretical grounding, and implementation of the study. The manuscript is well-developed, clear, and robust.
Below are my comments organized by section:
Abstract: it presents and describes the study in a concise and clear manner. Yet, it would be relevant to include some sociodemographic information about the families, such as the age of the parents and children. As for the keywords, they are appropriate. Nevertheless, their number exceeds typical recommendations and there appears to be some redundancy among them.
Introduction: it is very well written. It not only justifies the need for the study but also outlines the main processes and constructs involved, including the theoretical model that underpins the entire work. I suggest that when referring to systematic reviews (e.g., reference 17), the authors briefly mention the aim of those reviews to provide additional context.
Objectives: they are clear, well-grounded, and consistent with the introduction.
Method: this section is clearly explained and well-described. A few minor suggestions:
Lines 178–179 – Please add a citation for Bardin.
Lines 185–191 – Consider clarifying whether this section refers to the study design or procedures.
Lines 236–246 – Instead of describing the sections separately, why not integrate them directly into the explanation of the instrument’s structure?
Lines 314–315 – Was inter-rater agreement assessed? It would be appropriate to report the calculation of Fleiss' Kappa or the Intraclass Correlation Coefficient (ICC).
Results: they are well-structured and closely aligned with the method and study objectives.
Lines 336 and 344 – Please indicate the standard deviation (SD). Also, report the percentage distribution of parents' education levels.
Lines 345–346 – Report the SD for age and the percentages for the identified diagnoses.
Discussion and Conclusion: both sections are coherent and well-developed, with strong connections to the results and theoretical framework.
Once again, congratulations on this excellent work and it was a pleasure to review it.
Author Response
Response to Reviewer 3
Manuscript ID: healthcare-3698252
Title: Family Self-Care in the Context of Intellectual Disabilities: Insights from a Qualitative Study in Portugal
Comments and Suggestions Sent:
1 - Thank you very much for the opportunity to participate in the review of this work. It was a great pleasure for me to read and comment on this manuscript. Overall, I would like to congratulate the authors on the structure, theoretical grounding, and implementation of the study. The manuscript is well-developed, clear, and robust. Below are my comments organized by section.
Response: Thank you very much for your appreciation.
2 - Abstract: it presents and describes the study in a concise and clear manner. Yet, it would be relevant to include some sociodemographic information about the families, such as the age of the parents and children. As for the keywords, they are appropriate. Nevertheless, their number exceeds typical recommendations and there appears to be some redundancy among them.
Response: Thank you for your thoughtful feedback. We have revised the abstract to include essential sociodemographic details about the participating families, including the age ranges of parents and children. We also reviewed the keywords and reduced their number (despite Healthcare guidelines – “List three to ten pertinent keywords specific to the article yet reasonably common within the subject discipline”), eliminating redundancy while retaining relevance and discoverability.
3 - Introduction: it is very well written. It not only justifies the need for the study but also outlines the main processes and constructs involved, including the theoretical model that underpins the entire work. I suggest that when referring to systematic reviews (e.g., reference 17), the authors briefly mention the aim of those reviews to provide additional context.
Response: Thank you very much for your positive feedback. We appreciate your suggestion regarding reference 17 and other systematic reviews. We have revised the introduction to briefly include the aim or focus of each referenced systematic review, enhancing clarity and providing additional context for the reader. Also reference 21 was clarified on the introduction.
4 - Objectives: they are clear, well-grounded, and consistent with the introduction.
Response: Thank you for your feedback on the objectives.
5 - Method: this section is clearly explained and well-described. A few minor suggestions:
5.1 - Lines 178–179 – Please add a citation for Bardin.
Response: Thank you very much for your suggestion. We have added Bardin citation.
5.2 - Lines 185–191 – Consider clarifying whether this section refers to the study design or procedures.
Response: Thank you for this helpful observation. We have revised the text to clarify that this section refers specifically to data collection procedures. We have moved the content to the appropriate subsection (2.5. Interviews and data collection procedures) and adjusted the wording accordingly.
5.3 - Lines 236–246 – Instead of describing the sections separately, why not integrate them directly into the explanation of the instrument’s structure?
Response: Thank you for this helpful suggestion. We agree that integrating the description of the interview script into a single paragraph improves clarity and flow. We have revised the section accordingly to present the structure of the instrument more concisely and cohesively. The revised text now explains the five thematic sections of the interview guide in an integrated format, while maintaining the detail necessary to convey the breadth of topics addressed.
5.4 - Lines 314–315 – Was inter-rater agreement assessed? It would be appropriate to report the calculation of Fleiss' Kappa or the Intraclass Correlation Coefficient (ICC).
Response: Thank you for this valuable observation. While inter-rater agreement was not calculated using statistical measures such as Fleiss’ Kappa or the Intraclass Correlation Coefficient (ICC), we acknowledge the importance of transparency in analytic consistency. In this study, triangulation was conducted through collaborative coding and iterative discussion among three researchers (TDM, MJL, and EVC), following established qualitative rigor procedures. Our goal was to ensure conceptual clarity and consensus in theme development, rather than independent coding for statistical agreement. We have revised the manuscript to clarify this point.
6 - Results: they are well-structured and closely aligned with the method and study objectives.
6.1 - Lines 336 and 344 – Please indicate the standard deviation (SD). Also, report the percentage distribution of parents' education levels.
Response: Thank you for your insightful suggestion. We have now included the standard deviation for participants' age and added a percentage breakdown of parents' education levels to provide greater clarity and descriptive precision. These revisions have been incorporated into the demographic description in the Results section (lines 393-398).
6.2 - Lines 345–346 – Report the SD for age and the percentages for the identified diagnoses.
Response: Thank you for your suggestion. We have revised the manuscript to include the standard deviation (SD) for the children's ages and the percentage distribution of diagnoses, on the Results section (lines 410-414).
7 - Discussion and Conclusion: both sections are coherent and well-developed, with strong connections to the results and theoretical framework.
Response: Thank you for your comment.
8 - Once again, congratulations on this excellent work and it was a pleasure to review it.
Response: Thank you very much for your comment.

Reviewer 4 Report
Comments and Suggestions for Authors
Please see the attachment

The English is understandable, but contains long sentences, irregular verb tenses and Portuguese calques (e.g., “civil union”). I recommend professional academic editing in English and careful numbering of subsections, according to the Health template.
Author Response
Response to Reviewer 4
Manuscript ID: healthcare-3698252
Title: Family Self-Care in the Context of Intellectual Disabilities: Insights from a Qualitative Study in Portugal
Comments and Suggestions Sent:
- Introduction and Context
1.1. Explicit objective – The research goal can be inferred but is not stated directly. Add a clear sentence at the end of the introduction, for example: “…The objective of this study is to …”.
Response: Thank you for your helpful suggestion. We have now added an explicit sentence stating the research objective at the end of the introduction to enhance clarity and alignment with standard reporting practices.
1.2. Knowledge gap – The theoretical overview reads like an inventory of findings without a guiding thread. Reorganise it into thematic blocks (family quality of life, self-care, access barriers) and close each block with the specific gap your study addresses. Include recent reviews on family-centred care to underscore currency.
Response: Thank you for this helpful suggestion. In response, we have restructured the introduction into clearly defined thematic sections: (1) Family Quality of Life, (2) Self-Care and Family Self-Care (FSC), (3) Barriers to Access and Systemic Support, and (4) Study Aim and Relevance. Each section now concludes with a clearly articulated knowledge gap to ensure logical flow and theoretical coherence. We have also integrated recent systematic reviews on family-centered care, including Mestre et al. (2024) and Dunst et al. (2007) to underscore the relevance and currency of the topic. This reorganization enhances the clarity of our reasoning and better contextualizes the contribution of our study.
- Design and Methods
2.1. Methodological rationale – State explicitly which qualitative design you adopted (e.g., “descriptive-exploratory study with inductive content analysis”) and explain why it is suitable for investigating Family Self-Care. Clarify how it guided data collection and analysis.
Response: Thank you for your insightful comment. We have revised Section 2.1 (Design) to explicitly state the qualitative design used in the study. We now describe the research as a descriptive-exploratory qualitative study guided by an abductive analytical approach that integrates both inductive and deductive reasoning. We also provide a clear rationale for this choice, emphasizing its suitability for investigating the complex, context-dependent processes involved in Family Self-care. Additionally, we clarify how this design informed the development of the interview guide, the structure of data collection, and the application of thematic-categorical content analysis. These revisions aim to strengthen methodological transparency and address the criteria for qualitative rigor.
2.2. Participants and recruitment – Provide inclusion/exclusion criteria, recruitment sources, and the invitation/consent procedure. These details enhance transparency and reproducibility.
Response: Thank you for your helpful observation. We would like to clarify that the original manuscript did include the main aspects regarding participants and recruitment, including the inclusion criteria, sampling strategy, and interview participation details. However, in response to your comment, we have revised this section to make the following points more explicit:
- The source of recruitment (local primary healthcare services and the role of a community nurse).
- The invitation and informed consent procedure used prior to data collection.
- The addition of exclusion criteria to ensure transparency and reproducibility.
These revisions are now reflected in subsection 2.3. Participants and Inclusion/Exclusion Criteria to enhance clarity and methodological transparency. Thank you for helping us improve the precision of this section.
2.3. Analytic procedure – Indicate how many researchers coded the interviews, the consensus method used, and, if applicable, report an inter-rater coefficient (Kappa) or an external audit.
Response: Thank you for your valuable comment. In the revised manuscript, we have clarified the analytic process. Data analysis was conducted using Bardin’s thematic-categorical content analysis framework, supported by NVivo software. Three researchers (TDM, MJL, and EVC) independently coded the transcripts and subsequently engaged in iterative discussions to refine the coding scheme, resolve discrepancies, and reach consensus on the final categories and thematic structure. While no formal inter-rater reliability coefficient (e.g., Fleiss’ Kappa) was calculated, the use of investigator triangulation and peer debriefing enhanced the credibility and dependability of the analytical process. This multi-analyst approach ensured comprehensive interpretation grounded in both individual and collective researcher perspectives.
2.4. Qualitative rigour – The abbreviation list mentions the Consolidated Criteria for Reporting Qualitative Research (COREQ), yet it is not used in the text. Consider adding a brief table showing the techniques applied for credibility, transferability, dependability, and confirmability.
Response: Thank you for this valuable observation. Although we referenced the COREQ checklist, we acknowledge that its use was not made explicit in the manuscript. To address this, we have now incorporated a brief table summarizing the techniques used to ensure qualitative rigor across the four key dimensions: credibility, transferability, dependability, and confirmability (actual Table 1). This addition aligns with the COREQ guidelines and reinforces the transparency and trustworthiness of our study.
2.5. Ethics – Include the ethics-committee approval number (or clarify any exemption) and the date of approval, in line with Healthcare requirements.
Response: We appreciate the reviewer’s suggestion. The article already includes a dedicated section on ethics (Subsection 2.6), where the approval from the Ethics Committee is clearly stated. To enhance transparency and comply with Healthcare’s requirements, we have now added the ethics approval number and the date of approval directly in the text, as follows:
“This study received ethical clearance from the Ethics Committee of the University of Évora (reference number UE_22083) on November 16, 2022, and the Ethics Committee of the Local Health Unit of Lower Alentejo (reference number ULSBA/01/2023) on January 13, 2023, Portugal. Approval was given in accordance with the International Ethical Guidelines for Health-related Research Involving Humans.”
Thank you for helping us improve the clarity and completeness of this subsection.
- Results
3.1. Table 2 (FSC categories) – Categories and subcategories are listed, but please add the number of coded quotations for each subcategory (in parentheses or in a separate table). This will show data “density” and the relative weight of each theme without overloading the figure. These small improvements will sharpen the visual clarity of the Results section and help readers assess the strength of the evidence.
Response: Thank you for this constructive suggestion. We agree that including the number of coded quotations enhances transparency and allows readers to better assess the density and relative weight of each subcategory. We have revised previous Table 2 (now Table 3) by adding the number of coded quotations for each subcategory in a new column. This change improves the clarity and analytical rigor of the Results section without compromising visual simplicity.
- Discussion
4.1. Link to the literature – Incorporate recent international studies to contrast and contextualise your findings.
Response: We appreciate this valuable suggestion. In response, we have revised the Discussion section to incorporate and contrast our findings with recent international studies published between 2021 and 2025. Specifically, we integrated studies by Palm et al. (2021), Petrillo et al. (2022), Ismaili et al. (2024), Liaqat et al. (2025), and a review by Park et al. (2018) to strengthen the contextualization of our results.
These additions allow us to demonstrate how the experiences of families in Portugal align with broader international trends—such as emotional resilience, informal learning, systemic access barriers, and the value of collaborative care. This helps underscore both the global relevance and the context-specific nuances of the FSC framework.
Please see the revised Discussion section, particularly paragraphs in the Physical, Cognitive, and Psychosocial domains.
4.2. Limitations – Reflect on the geographic restriction (central-northern Portugal), possible social-desirability bias, and the small sample size.
Response: Thank you for your insightful observation. We acknowledge the importance of clearly outlining study limitations to enhance transparency and interpretability. While the manuscript already mentioned the sample size and geographic scope, we have now revised the Limitations section to explicitly address: The geographic restriction to the Lower Alentejo region and its implications for transferability; The potential for social desirability bias, especially given the involvement of a community nurse and the sensitive nature of the topic; The impact of the small sample size and reliance on self-reported data. These clarifications have been incorporated to strengthen the discussion and more accurately contextualize the findings (please see Subsection 4.6. Strengths and Limitations).
4.3. Implications – Explain how the FSC framework can guide family-education programmes, community-nursing protocols, and disability-support policies.
Response: We appreciate this valuable suggestion. In response, we have expanded Discussion Section to explicitly outline how the FSC framework - comprising physical, cognitive, psychosocial, and behavioral domains - can guide practical applications in clinical, community, and policy settings. Specifically, we describe how the framework can inform the development of structured family education programs, shape community-nursing protocols that focus on whole-family well-being, and support the integration of family self-care into disability-support policies. These additions enhance the translational relevance of our findings and clarify how FSC can be operationalized in diverse care contexts (please see Subsection 4.5. Implications for Practice, Policy, and Research).
- Conclusions
Temper the generalisation of results, stress their exploratory nature, and suggest future quantitative studies to validate the FSC framework in other contexts..
Response: We thank the reviewer for this important observation. In response, we have revised the conclusion to explicitly underscore the exploratory nature of the study and the limited generalizability of the findings due to the small sample size and regional context. We have also added a recommendation for future research to include quantitative and mixed-methods designs aimed at validating and expanding the FSC framework across diverse cultural and healthcare settings.
- References
Several entries do not follow the journal format—for instance, in reference 13 the initials precede surnames. Apply the MDPI/ACS style consistently (authors, title, journal in italics, year, volume, pages, DOI).
Response: Thank you for your observation. We have carefully reviewed and revised all reference entries to ensure full consistency with the Healthcare journal style guidelines. This includes correcting the order of author names (surname followed by initials), italicizing journal titles, and standardizing the format for year, volume, page numbers, and DOI as the journal's requirements.
- Language and Style
The English is understandable but contains long sentences, irregular verb tenses, and Portuguese calques (e.g., “civil partnership”). I recommend professional academic-English editing and careful numbering of subsections in accordance with the Healthcare template.
Response: We appreciate the reviewer’s helpful comments regarding language and style. In response, we have thoroughly revised the manuscript to improve sentence structure, correct irregular verb tenses, and eliminate calques and other non-native constructions. Particular attention was given to improving clarity and flow throughout the text. Additionally, subsection numbering has been reviewed and aligned with the Healthcare journal template. We trust that these revisions have significantly enhanced the readability and overall presentation of the manuscript.

Round 2
Reviewer 2 Report
Comments and Suggestions for Authors
Thank you for taken the time to revise your paper and address the feedback. I feel you have a more robust paper as a result and wish you the best with any further work in this area.
Author Response
We sincerely thank the reviewer for the thoughtful feedback and kind words. We truly appreciate the time and care invested in reviewing our work. Your comments were instrumental in helping us strengthen the manuscript, and we are grateful for your support. We look forward to continuing research in this area and hope the findings contribute meaningfully to the field.
Reviewer 4 Report
Comments and Suggestions for Authors
Please see the attachment.

Although the prose has improved, very long sentences and some Portuguese/English accents persist (e.g., "civil union", "care unit"). A final professional language correction would improve clarity.
Author Response
Response to Reviewer 4 (Round 2)
Manuscript ID: healthcare-3698252
Title: Family Self-Care in the Context of Intellectual Disabilities: Insights from a Qualitative Study in Portugal
Comments and Suggestions Sent:
- Discussion
- Limitations – Add an explicit subsection covering at least: small sample size, specific geographical setting, possible social-desirability bias in the interviews, and limited transferability to other sociocultural contexts.
Response: Thank you for your recommendation. In the revised version, we have expanded the Limitations subsection within the Discussion section. We now explicitly address the small sample size (n = 9 families), the specific geographical context of the Lower Alentejo region in Portugal, the potential for social desirability bias due to a perceived association with health professionals, and the limited transferability of findings to other sociocultural and healthcare contexts. These revisions are highlighted in green in the revised manuscript. They aim to provide a clearer understanding of the scope and limitations of our study’s findings.
- Practical and policy implications – Expand with concrete examples of how your findings could inform community-nursing protocols, caregiver-training programmes, or municipal respite initiatives.
Response: We appreciate this suggestion. The Implications for Practice, Policy, and Research section has been expanded to include concrete examples. Specifically, we describe how the FSC framework can inform community-nursing protocols by promoting whole-family assessments and culturally sensitive care planning; support caregiver-training programmes by guiding tailored education based on the FSC domains; and inspire municipal respite initiatives by advocating for policies that recognize caregivers as essential partners and ensure access to supportive services. These revisions are highlighted in green in the revised manuscript.
- Conclusions – Temper any claim of generalisability and include a clear statement on the need for future (quantitative or mixed-methods) studies to validate and extend the Family Self-Care framework.
Response: Thank you for this insightful observation. The Conclusions section has been revised to temper claims of generalisability by stating that the findings are exploratory and context-specific. We now highlight the limitations due to the small sample size and regional scope and explicitly call for future quantitative and mixed-methods studies to validate, refine, and extend the FSC framework across different sociocultural and healthcare contexts. These revisions are highlighted in green in the revised manuscript.
- References and formatting
Inconsistencies with MDPI style remain (order of initials and surnames, use of italics, DOI format). Please review the entire reference list carefully.
Response: Thank you for your observation. In the revised manuscript, we have carefully reviewed and corrected the entire reference list to ensure full compliance with MDPI formatting guidelines. This includes standardizing the order of author initials and surnames, applying the appropriate use of italics (e.g., journal names), and formatting all DOIs as active links without italics. These adjustments have been implemented consistently throughout the reference section to meet the journal’s style requirements.
- Language
Although the prose has improved, very long sentences and some Portuguese/English calques persist (e.g., “civil partnership,” “caregiving unit”). A final professional language edit would enhance clarity.
Response: We appreciate the reviewer’s careful attention to language clarity. We confirm that the expression “civil partnership” or “caregiving unit” does not appear in the current version of the manuscript. However, we acknowledge the broader point regarding potential calques and overly long sentences. To address this, we conducted a comprehensive language revision and consulted a native English-speaking academic editor with experience in qualitative health research. The manuscript was carefully reviewed to improve sentence structure, eliminate any residual calques, and enhance overall readability.
